# Rotavirus NSP1 Inhibits Type I and Type III Interferon Induction

**DOI:** 10.3390/v13040589

**Published:** 2021-03-31

**Authors:** Gennaro Iaconis, Ben Jackson, Kay Childs, Mark Boyce, Stephen Goodbourn, Neil Blake, Miren Iturriza-Gomara, Julian Seago

**Affiliations:** 1The Pirbright Institute, Ash Road, Woking, Surrey GU24 0NF, UK; gi239@cam.ac.uk (G.I.); ben.jackson@pirbright.ac.uk (B.J.); kay.childs@pirbright.ac.uk (K.C.); 2Institute of Infection, Veterinary and Ecological Sciences, University of Liverpool, Liverpool L69 7BE, UK; nwblake@liverpool.ac.uk (N.B.); miturrizagomara@path.org (M.I.-G.); 3Division of Structural Biology, Wellcome Trust Centre for Human Genetics, University of Oxford, Oxford OX3 7BN, UK; mark.boyce@strubi.ox.ac.uk; 4Institute for Infection and Immunity, St. George’s, University of London, London SW17 0RE, UK; goodbour@sgul.ac.uk

**Keywords:** rotavirus, type I interferon, type III interferon, NSP1, IRF-1, IRF-3, IRF-7, NF-κB

## Abstract

Type I interferons (IFNs) are produced by most cells in response to virus infection and stimulate a program of anti-viral gene expression in neighboring cells to suppress virus replication. Type III IFNs have similar properties, however their effects are limited to epithelial cells at mucosal surfaces due to restricted expression of the type III IFN receptor. Rotavirus (RV) replicates in intestinal epithelial cells that respond predominantly to type III IFNs, and it has been shown that type III rather than type I IFNs are important for controlling RV infections in vivo. The RV NSP1 protein antagonizes the host type I IFN response by targeting IRF-3, IRF-5, IRF-7, or β-TrCP for proteasome-mediated degradation in a strain-specific manner. Here we provide the first demonstration that NSP1 proteins from several human and animal RV strains antagonize type III as well as type I IFN induction. We also show that NSP1 is a potent inhibitor of IRF-1, a previously undescribed property of NSP1 which is conserved among human and animal RVs. Interestingly, all NSP1 proteins were substantially more effective inhibitors of IRF-1 than either IRF-3 or IRF-7 which has significance for evasion of basal anti-viral immunity and type III IFN induction in the intestinal epithelium.

## 1. Introduction

Rotaviruses (RVs) are members of the *Rotavirus* genus, one of 16 genera of the Reoviridae family that are grouped into at least 9 distinct species, named RVA-RVJ (reviewed in [1]). Group A RVs (RVAs) are a leading cause of infectious gastroenteritis in infants and are responsible for more than 100,000 deaths/year in children under 5 years of age, predominantly in low-income countries [2]. The young of many other mammalian species are also susceptible to infection by distinct RVA strains that exhibit a predominantly host-specific distribution pattern. Experiments in suckling mice showed that animals infected with a murine RVA strain exhibit clinical signs even following a low infectious dose, whereas simian, bovine, or porcine RVA strains replicate poorly in the heterologous murine host and fail to induce disease [3,4]. Although the molecular basis of the observed host-range restriction remains poorly understood, the host interferon (IFN) system is known to be an important factor in controlling replication of heterologous RV strains in the mouse model [5].

The production of type I and type III IFNs by infected cells is one of the earliest responses of the host to virus infection. IFNs are secreted proteins with potent anti-viral activity that are critically important for controlling virus replication prior to the development of an adaptive immune response (reviewed in [6]). In humans, the type I IFNs consist of multiple subtypes, with the 13 IFN-α and single IFN-β subtypes being the most relevant for virus infection. Their biological effects are mediated through dimeric type I IFN receptors, which are found on virtually all nucleated cells. The type III IFNs, IFN-λ1 (IL-29), IFN-λ2 (IL-28A), and IFN-λ3 (IL-28B), are produced by many cell types, but act exclusively on epithelial cells which express the IFN-λ receptor subunit IFNLR1. Consequently, type III IFNs have a specific role in suppressing virus replication at barrier surfaces such as the respiratory and gastrointestinal mucosa (reviewed in [7]). In vivo, RVs replicate in mature enterocytes at the tip of the villi lining the small intestine, and these intestinal epithelial cells (IECs) produce both type I and type III IFNs following RV infection in mice [8]. However, whilst IECs express high levels of both IFN-λ receptor subunits, they express comparatively low levels of the IFN-α/β receptor subunits, and hence mount a stronger response to type III, than to type I IFNs [9], a feature that was also observed in RV-infected human intestinal enteroids (HIEs) [10]. Furthermore, studies in IFN receptor knockout mice showed that mice lacking IFN-λ receptors are far more susceptible to RV infection, and develop more severe pathology following infection, than wild-type (WT) mice or mice lacking type I IFN receptors [8]. These data suggest that type III IFNs are more important for controlling RV infections in vivo, although type I IFNs may play a role in preventing systemic infection beyond the epithelial barrier [9], and interestingly, exogenous type I IFN was more effective than type III in controlling RV infection of HIEs [10].

The trigger for production of type I and III IFNs is the recognition of viral components termed pathogen-associated molecular patterns (PAMPs) by host pattern recognition receptors (PRRs). RVs replicate in the cytoplasm and are sensed by the RLRs RIG-I and MDA5, which are activated by binding to dsRNA molecules produced during virus replication [11,12]. This allows their association with the mitochondrial antiviral signaling protein (MAVS), which in turn recruits the kinases TANK-binding kinase 1 (TBK1) and IκB kinase (IKK) ε (TBK1/IKKε), and IKKα/β/γ, that activate the transcription factors IFN regulatory factor 3 (IRF-3) and NF-κB respectively. IRF-3 is constitutively expressed in most cell types but remains inactive in the cytoplasm until it is phosphorylated by TBK1/IKKε, enabling dimerization and nuclear translocation. The NF-κB heterodimer p50/p65 is also cytoplasmic in unstimulated cells, where it is sequestered by the inhibitor protein IκB. Phosphorylation of IκB by the IKK complex creates a binding site for an E3 ubiquitin ligase, which targets IκB for ubiquitination and degradation via the proteasome releasing NF-κB to move into the nucleus and co-operate with IRF-3 to induce transcription of IFN genes. The IFN-β and IFN-λ1/2/3 promoters contain binding sites for IRF-3 and NF-κB and are directly inducible by virus infection through this pathway, whereas in most non-immune cells, IFN-α production is a secondary response that follows autocrine stimulation by IFN-β and upregulation of IRF-7. Mitochondrial MAVS can induce both IFN-β and IFN-λ expression, peroxisomal MAVS specifically signals production of IFN-λ in a manner that depends upon IRF-1 [13], however subsequent data have contradicted this [14]. Secretion of IFNs leads to autocrine and paracrine signaling through type I and type III IFN receptors on the surface of neighboring cells. Although structurally unrelated, engagement of these receptors by their respective ligands initiates similar intracellular responses through activation of the Janus kinase/signal transducer and activator of transcription (Jak/STAT) signal transduction pathway [15]. Phosphorylated STAT1 and STAT2 bind IRF-9 to form ISGF3, that translocates to the nucleus and activates transcription of hundreds of IFN-stimulated genes (ISGs).

For virus replication to proceed, viruses must employ effective evasion strategies to disrupt the signaling pathways leading to IFN production and/or ISG expression (reviewed in [6]). For RVs, this is a function of the non-structural protein NSP1, which targets several host proteins involved in the IFN response for proteasome-mediated degradation in a strain-specific manner (reviewed in [16]). NSP1 has a conserved N-terminal RING domain, and a highly variable C-terminus which is responsible for interacting with cellular targets. The divergent properties of NSP1 proteins from different strains may play a role in determining host range restriction if NSP1 fails to counteract the IFN response in a heterologous host. For example, the simian RVA strain, RRV, grows poorly in mice when compared to murine strains due to the sensitivity of this virus to the murine IFN system [5], and this has been linked to differences in the NSP1 gene [4].

Members of the IRF family, including IRF-3, IRF-5, IRF-7, and IRF-9, are important targets of RVA NSP1 [17,18,19,20]. NSP1 from several bovine, murine, and simian RVA strains interact directly with IRF-3 and promotes its degradation via the proteasome, thus preventing transcription of IFN-β [17,18,21]. Interestingly, the NSP1 proteins from two bovine strains could degrade human, but not murine IRF-3, suggesting that species differences in IRF-3 can affect the ability of some strains to inhibit IFN induction in a particular host [22]. Notably, the NSP1 proteins from most human RVA strains are unable to degrade human IRF-3, but alternatively target IRF-5 and IRF-7 [23]. Furthermore, NSP1 from the porcine RV strain OSU completely lacks the ability to degrade IRFs, but instead prevents activation of NF-κB by promoting proteasomal degradation of β-transducin repeat containing protein (β-TrCP) [24], a subunit of the SkpI/CulI/F-box complex SCF^β-TrCP^ which recognizes the phosphorylated form of IκB and targets it for degradation [25]. The loss of β-TrCP in RV-infected cells leads to the stabilization of IκB and the retention of NF-κB in cytoplasmic complexes. The ability to degrade β-TrCP is shared by many human RVA strains which are unable to degrade IRF-3 [26]. The binding sites for IRF-3 and β-TrCP overlap at the extreme C-terminus of NSP1 [17,26,27]. NSP1s that bind IRF-3 contain the motif pLxIS (where p is a hydrophilic residue and x is any amino acid) also found in MAVS, TRIF, and STING, the adapter proteins downstream of RLRs, TLRs, and the DNA sensor cGAS respectively, that recruit IRF-3 into signaling complexes [28]. In the case of MAVS, TRIF, and STING, the serine residue in this motif must be phosphorylated in order to interact with IRF-3 [29], however phosphorylation of NSP1 is not required, and other residues outside of the core motif additionally contribute to IRF-3 binding [28]. At the C-terminus of NSP1 proteins which target β-TrCP, the sequence DSGIS exists in place of pLxIS (Figure 1B) [26,27]. DSGIS resembles the phosphodegron motif (DSGΦxS, where Φ represents a hydrophobic residue) present in IκB which, when phosphorylated on both serines by IKK, mediates the interaction with β-TrCP. In contrast to the NSP1 interaction with IRF-3, the interaction with β-TrCP does require phosphorylation of the serines within DSGIS, however the kinase responsible for the phosphorylation of NSP1 is casein kinase II (CKII) and not IKKβ [30].

Other known targets of NSP1 that form part of the IFN induction pathway include RIG-I [31], MAVS [32] and TRAF2 [33]. Disruption of Jak/STAT signaling downstream of the type I IFN receptor is also achieved by NSP1 through inhibition of STAT1/2 phosphorylation and nuclear import [34,35]. The sum of these interactions is likely to have a critical impact on the ability of the host to control virus replication.

Here, we investigated whether RV NSP1 proteins inhibit the production of type III in addition to type I IFNs using a selection of well characterized and previously untested NSP1 proteins from human and animal RV strains. We demonstrate that activation of type III IFN induction by virus infection is inhibited by human and animal RV NSP1s, and reveal a previously undescribed role for NSP1 as an inhibitor of IRF-1.

## 2. Materials and Methods

### 2.1. Cells and Virus

Human embryonic kidney (HEK)-293 cells (ATCC**^®^** CRL-1573™) were obtained from ATCC and maintained in DMEM (Thermo Fisher Scientific, Paisley, UK #41965-039) supplemented with 5% fetal bovine serum (FBS) (Thermo Fisher Scientific, #10270106), penicillin (100 U/mL) and streptomycin (100 μg/mL) (Sigma, St. Louis, MO, USA #P4333). Cell passage numbers ranged from 20 to 50. Sendai virus (SeV), Cantell strain (ATCC VR-907 Parainfluenza 1) was purchased from Charles River, Wilmington, NC, USA (#10100773) at a titre of 4000 hemaglutination (HA) units/mL.

### 2.2. Plasmids

DNA sequences containing the open reading frames (ORFs) encoding NSP1 proteins from the human RVA strains RG177/13 (accession number (AN): Ku048673; herein referred to as RG177), BID2QJ (AN: MG181761), and Wa (AN: JX406751), and the porcine RVA strains ROTA04 (AN: KJ482247) and P343 (AN: AB972862), were synthesised *de novo* by GeneArt gene synthesis (Thermo Fisher Scientific) and cloned into pcDNA3.1. Plasmids containing the cDNAs encoding the NSP1 proteins from the bovine RVA strain UKtc (AN: KC215492), the simian RVA strain RRV (AN: HQ846847), and the porcine RVA isolate OSU (AN: D38153) were provided by Prof Malcolm McCrae (University of Warwick, Coventry, UK). For expression in yeast as GAL4 DNA-binding domain fusion proteins (“baits”), DNA fragments containing NSP1 ORFs were amplified from these plasmids by polymerase chain reaction (PCR) using primers containing appropriate restriction enzyme recognition sequences, and subsequently cloned into pGBKT7 (Takara, Saint-Germain-en-Laye, France). Similarly, for expression in mammalian cells, ORFs encoding RVA NSP1s were amplified from the respective pGBKT7 plasmids by PCR and cloned into pcDNA3.1/V5-His-TOPO**^®^** (Thermo Fisher Scientific).

The cDNA encoding human IRF-3 was a kind gift from Takashi Fujita (Kyoto University, Kyoto, Japan), the cDNA encoding porcine IRF-3 (AN: NM_213770) was synthesised *de novo* and cloned into pcDNA3.1 by GeneArt gene synthesis (Thermo Fisher Scientific), and the cDNA encoding bovine IRF-3 was prepared from calf testes cells using standard methods. For expression in yeast as GAL4 activation domain fusion proteins (“preys”), cDNAs encoding IRF-3 proteins were amplified by PCR using primers containing appropriate restriction enzyme recognition sequences and cloned into pGADT7 (Clontech). The plasmid pGADT7.sIRF-3 encoding simian IRF-3 was a kind gift from Prof Michele Hardy (Washington State University, Pullman, WA, USA).

The plasmids pcDNA3.1.CSFV-N^pro^ encoding the N^pro^ protein from the Alfort strain of classical swine fever virus (CSFV) [36], the IRF-1 expression vector pEF.IRF-1 [37] and the plasmid pRc/CMV.p65 for expression of the p65 subunit of NF-κB [38] have been described elsewhere. The plasmids pRC-IKKβ and pMC159 encoding IKKβ and the MC159 protein of molluscum contagiosum virus (MCV) respectively were kind gifts from Prof Johanna Shisler (Department of Microbiology, University of Illinois Urbana-Champaign, Urbana, IL, USA). The plasmid pcDNAV5.RPV_Sa_VCStop encoding the V protein of rinderpest virus (RPV, Saudi/81 strain) was a kind gift from Dr. Michael Baron (The Pirbright Institute, Surrey, UK). The plasmid pEGFP-C1-IRF-7 was a kind gift from Prof John Hiscott (Laboratorio Pasteur, Istituto Pasteur-Fondazione Cenci Bolognetti Rome, Rome, Italy). The plasmid encoding the constitutively active form of IRF-3, pEF.FLAG.IRF-3-5D, was generated by performing site-directed mutagenesis to introduce the following amino acid changes into the human IRF-3 cDNA: S396D, S398D, S402D, T404D, S405D. The identity of all new cDNA constructs was confirmed by DNA sequencing.

The firefly luciferase reporter gene constructs under the control of the intact human IFN-β promoter (pIFΔ(-116)lucter; [39]) or 5 copies of the NF-κB-responsive PRDII element of the IFN-β promoter (p(PRDII)_5_tkΔ(-39)luc) [40] have been described elsewhere. The firefly luciferase reporter plasmids under the control of the human IFNλ1, IFNλ3 or IFNα4 promoters (pGL2/hIFNλ1, pGL2/hIFNλ3, and pGL2/hIFNα4) were kind gifts from Dr Pamela Osterlund (National Institute for Health and Welfare, Helsinki, Finland), the firefly luciferase reporter plasmid under the control of the murine Mx1 promoter (pGL3-Mx1P-luc) was a kind gift from Prof Georg Kochs (University of Freiburg, Freiburg im Breisgau, Germany), and the firefly luciferase reporter plasmid under the control of the IRF-responsive PRDIII-I element of the IFN-β promoter (PRD(III-I)-Luc) was a kind gift from Prof Stephan Ludwig, (University of Muenster, Germany). The plasmid directing constitutive expression of renilla luciferase, pRL-CMV, was obtained from Promega.

### 2.3. Yeast Two-Hybrid

Combinations of bait (pGBKT7) and prey (pGADT7) plasmids were introduced into yeast strain AH109 (Clontech) using PEG/LiAc mediated transformation and plated onto synthetic dropout (SD) medium lacking tryptophan and leucine (SD-Trp-Leu) to select for the successful uptake of both plasmids. Plates were incubated for 5 days at 30 °C to obtain yeast colonies, which were then re-streaked onto triple dropout (TDO) media additionally lacking histidine (SD-Trp-Leu-His) and quadruple dropout (QDO) media additionally lacking histidine and adenine (SD-Trp-Leu-His-Ade) to assay for protein-protein interactions. Growth of yeast on TDO and QDO media was monitored for 5 days at 30 °C and scored as − (no growth), + (growth on TDO but not QDO) or ++ (growth on QDO).

### 2.4. Luciferase Assays

HEK-293 cells were plated at 0.25 × 10^6^ cells/well in 24-well plates and the following day transfected with mixtures of plasmids using TransIT**^®^**-LT1 transfection reagent (Mirus Bio, Madison, USA). Plasmid mixtures contained a firefly luciferase reporter plasmid, the pRL-CMV plasmid directing constitutive expression of renilla luciferase as an internal transfection control, and either the empty pcDNA3.1 plasmid or pcDNA3.1 encoding RV NSP1 or other viral proteins used as controls. Where required, additional plasmids directing expression of IRF-3-5D, IRF-7, IRF-1, IKKβ, β-TrCP or p65 were included; 24 h after transfection, cells were infected with SeV or treated with Tumour Necrosis Factor (TNF) α (5 ng/mL) or porcine IFNα (500 U/mL) where indicated; 40 h after transfection, cell extracts were prepared and measurements of firefly and renilla luciferase activity were taken using the Dual-Luciferase**^®^** Reporter Assay System (Promega, Chilworth, UK). Firefly luciferase values were normalized to the renilla luciferase values obtained from the same sample, and within each experiment the induced sample containing the empty vector was given a value of 100%. For all experiments, the data presented are the means of 3 readings obtained from 3 independent experiments conducted on separate occasions, with the error bars representing the standard deviation.

### 2.5. Statistical Analyses

Statistical analyses were carried out using Graph Pad PRISM 7.00. Data are presented as mean ± standard deviation (SD). An unpaired Student’s *t*-test was used to compare two groups. In all figures, the statistical significance between the indicated samples and the induced control is designated as * *p* < 0.05, ** *p* < 0.01, *** *p* < 0.001, **** *p* < 0.0001, or NS (*p* > 0.05).

## 3. Results

### 3.1. Non-Structural Proteins (NSP1s) from Human and Animal Group A Rotavirus (RVA) Strains Inhibit Interferon (IFN)-β Induction in Human Cells

First, we compared the ability of NSP1 proteins from a selection of human and animal RVA strains to inhibit SeV activation of a luciferase reporter gene under the control of the IFN-β promoter in human cells. SeV infection resulted in strong activation of luciferase activity in the presence of the empty vector pcDNA3 (Figure 1A, black bars). The NSP1 proteins from the three human RVA strains reduced activation of the IFN-β promoter by 35–50%, but this property was not restricted to the human RVA NSP1s, since, with the exception of the porcine ROTA04 NSP1 which had no effect, the animal RVA NSP1s also reduced IFN-β induction to a similar or a greater extent (Figure 1A, colored bars). The N^pro^ protein from CSFV, which inhibits IFN-β induction by targeting IRF-3 for degradation [36,41], served as a positive control.

### 3.2. NSP1 Proteins Inhibit Activation of the IFN-β Promoter by IFN Regulatory Factor (IRF)-3

Induction of the IFN-β promoter in response to SeV depends on both IRF-3 and NF-κB, and although RVA NSP1s exhibit a conserved ability to suppress IFN-β induction, this may be through inhibition of either IRF-3 or NF-κB activity. The NSP1s from P343, UKtc, and RRV contain the IRF-3-binding pLxIS motif, whilst the NSP1s from the human RVAs all contain the β-TrCP-binding DSGIS sequence that is present in OSU NSP1 (Figure 1B). The NSP1 gene of ROTA04 is of the A8 genotype which clusters phylogenetically with the genotype A1 and A2 NSP1s found predominantly in human and porcine RV strains [27]. However, whilst 98% of these “OSU-like” NSP1s contain the DSGIS motif, ROTA04 NSP1 contains DLSMS, which is a slightly better match to the IRF-3-binding motif due to the leucine at the second position, but has a methionine at position 4 instead of isoleucine or leucine. Since ROTA04 was unable to inhibit activation of the IFN-β promoter (Figure 1A), this C-terminal sequence may be suboptimal for inhibition of either transcription factor. From analysis of the C-terminal sequences, we would predict that the human RVA NSP1s would inhibit IFN-β induction by blocking NF-κB activation, and the animal RVA NSP1s would inhibit IRF-3. To investigate this, we first examined the effects of NSP1 proteins on activation of a luciferase reporter gene under the control of the IRF-3-responsive element of the IFN-β promoter, PRDIII-I. Surprisingly, the NSP1s from the human as well as the animal RVAs, including ROTA04, all inhibited activation of PRDIII-I in response to SeV (Figure 1C), although the P343, UKtc and RRV NSP1s resulted in the greatest reductions in promoter activation.

The PRDIII-I region of the IFN-β promoter can bind several members of the IRF family of transcription factors, so in order to examine the effects of NSP1 proteins on IRF-3 directly, we activated the PRDIII-I reporter by transfection of a plasmid encoding a constitutively active form of IRF-3, IRF-3-5D, in which 5 inducibly phosphorylated serine/threonine residues are changed to phosphomimetic aspartic acid residues. All NSP1 proteins tested inhibited activation of the PRDIII-I reporter by IRF-3-5D, although P343, UKtc and RRV NSP1s were reproducibly more effective inhibitors than ROTA04 or the three NSP1s from human RVA strains (Figure 1D). These data show that although human RVA NSP1 proteins would be expected to be inhibitors of NF-κB rather than IRF-3 based on their C-terminal sequences, they do have some capacity to inhibit IRF-3 activity, albeit to a lesser degree than the NSP1s that contain the pLxIS motif.

Next, we used the yeast two-hybrid assay to investigate whether the human RVA NSP1s interact with human IRF-3. Also, since species differences in IRF-3 proteins are responsible for the failure of NSP1s from certain bovine strains to degrade IRF-3 in murine cells [22], we compared the ability of NSP1s from different strains to interact with IRF-3 from homologous or heterologous hosts. Yeast cells transformed with plasmids encoding NSP1s as GAL4-DNA binding domain (DBD) fusions (baits) and human (h), simian (s), porcine (p) or bovine (b) IRF-3 as GAL4-activation domain (AD) fusions (preys) were streaked onto triple drop-out (TDO) and quadruple drop-out (QDO) media to assay for bait–prey interactions. Growth on TDO medium relies on activation of the HIS3 reporter gene, whilst growth on QDO medium requires activation of both the HIS3 and the ADE2 reporter genes and is, therefore, a higher stringency selection. Interestingly, RG177 and BID2QJ NSP1 proteins were able to interact with hIRF-3 despite the absence of the pLxIS motif (Figure 1E). P343 and UKtc NSP1 proteins also interacted with hIRF-3, whereas Wa and RRV NSP1s did not, although they displayed a similar ability to inhibit hIRF-3 in the luciferase assay. It was not possible to use the ROTA04 NSP1 in the yeast two-hybrid system due to prey-independent transactivation of the reporter genes by this protein as a GAL4-DBD fusion. The yeast expressing P343 and UKtc NSP1s with IRF-3 were the only transformants to grow on the QDO media (Figure 1E), suggesting that these interactions are stronger than the interactions between the other NSP1s and IRF-3. This correlates with the greater effectiveness of these NSP1 proteins at inhibiting IRF-3 in the luciferase assays compared to the human RVA NSP1s, however this correlation does not hold for RRV NSP1 which is equally as effective as P343 and UKtc in the luciferase assays but does not interact with IRF-3 in yeast. The failure to detect an interaction between hIRF-3 and RRV NSP1 was not due to species differences since RRV NSP1 did not bind sIRF-3 either. Surprisingly, Wa NSP1 interacted with sIRF-3 and bIRF-3 suggesting that it does have IRF-3 binding activity, but perhaps forms a weaker interaction that is harder to detect. Overall, there is no species specificity with regards to the ability of RVA NSP1s to target IRF-3, since the NSP1 proteins that interact with IRF-3 from the homologous species, also interact similarly with IRF-3 from other species.

### 3.3. Inhibition of IFN-α4 Production by RVA NSP1

Several human RVA NSP1s, including Wa, that are unable to cause IRF-3 degradation, instead target IRF-7 through an equivalent interaction with the conserved C-terminal IRF-association domain (IAD) present in both IRFs [19,20,23]. To determine whether the NSP1 proteins used here can inhibit IRF-7 activity, we used a luciferase reporter gene under the control of the IFN-α4 promoter which depends on activation of IRF-7, rather than IRF-3, for virus-inducible transcription. Since HEK-293 cells do not produce IRF-7, this reporter gene is not inducible by SeV in these cells, however it can be activated by transfection of a plasmid encoding IRF-7 (Figure 2A, black bars). In this assay, only expression of UKtc NSP1 or the positive control, CSFV N^pro^ [42], reduced activation of the IFN-α4 reporter, whilst the other NSP1s had no effect (Figure 2A). Since IRF-7 is the direct target of NSP1, we considered whether excess production of IRF-7 might overcome the capacity of NSP1 to inhibit it. Therefore, we repeated the experiment with a smaller amount of the plasmid encoding IRF-7, which is insufficient to activate the IFN-α4 promoter on its own but does enable activation in response to SeV (Figure 2B, black bars). Under these conditions, RG177, Wa, and P343 NSP1s reduced IFN-α4 induction by 30-40% and UKtc NSP1 was most effective, reducing induction by 60% (Figure 2B). ROTA04 and RRV NSP1s did not inhibit IFN-α4 induction, implying that they are unable to block IRF-7 activity.

### 3.4. Inhibition of NF-κB Activation by RVA NSP1

To investigate the effects of NSP1 proteins on NF-κB activation, we used a luciferase reporter gene under the control of the NF-κB binding site from the IFN-β promoter (PRDII), and stimulated cells with TNFα (Figure 3A). As controls we used two known inhibitors of NF-κB activation, the MC159 protein from MCV [43], and the NSP1 protein from the porcine RV strain OSU [24]. Both control proteins reduced the level of NF-κB activation in response to TNFα as expected (Figure 3A, grey bars). Furthermore, all RVA NSP1 proteins tested with both types of C-terminal motif also inhibited NF-κB activation (Figure 3A, colored bars). Interestingly, UKtc NSP1 which contains the IRF-3 binding pLxIS motif rather than DSGIS was the most effective inhibitor of NF-κB.

To investigate the mechanism by which NSP1 proteins inhibit NF-κB activation, and to determine whether this is through targeting β-TrCP or something else, we activated the NF-κB-responsive reporter gene by overexpression of different components of the NF-κB pathway. Many different extra- and intra-cellular signals including virus infection and TNFα can lead to activation of the canonical NF-κB pathway, and although the upstream receptors and receptor-associated proteins may be unique to each signal, all pathways converge on the IKK complex. To determine whether NSP1 acts downstream of IKK we overexpressed IKKβ, which resulted in a 2-fold increase in promoter activity (Figure 3B, black bars). Although this increase is small, we did observe a reduction in NF-κB activation in the presence of all the NSP1 proteins, and again we observed that the OSU and UKtc NSP1s were the most effective (Figure 3B). This shows that the target of NSP1 lies downstream of IKKβ. Next, we overexpressed β-TrCP, this also activated NF-κB (Figure 3C, black bars), and all NSP1 proteins inhibited this to a similar extent (Figure 3C). Finally, we overexpressed the p65 subunit of NF-κB, and in this experiment RG177, BID2QJ, Wa, P343 and ROTA04 NSP1 failed to inhibit NF-κB activation, indicating that these NSP1s interfere with NF-κB activation upstream of p65 (Figure 3D). For OSU, UKtc and RRV NSP1s, a small reduction of NF-κB activation in response to p65 was still observed, although this is not as significant a reduction as that observed when TNFα, IKKβ or β-TrCP were used as activators of NF-κB. These data show that the ability to interfere with NF-κB activation is a conserved feature across all NSP1s tested, irrespective of the presence of the DSGIS motif. Inhibition occurs between the phosphorylation of IκB by IKKβ and release of NF-κB from IκB, which would be consistent with β-TrCP being the target of NSP1. Additionally, OSU, UKtc and RRV NSP1s may have some ability to inhibit p65 activity directly.

### 3.5. Inhibition of Type III IFN Induction by RVA NSP1

Like IFN-β, the type III IFNs are directly induced by virus infection, and are activated by members of the IRF family and NF-κB [44,45]. Therefore, it seemed likely that RV NSP1 proteins would block induction of type III IFNs in addition to type I IFNs, which may be critical to countering the innate immune response in the intestinal epithelium. To investigate this possibility, luciferase reporter genes under the control of either the IFN-λ1 or the IFN-λ3 promoters were transfected into cells and induced by infection with SeV. Activation of the IFN-λ1 promoter by SeV was substantially reduced in cells expressing all RVA NSP1 proteins except for ROTA04 (Figure 4A). Except for UKtc NSP1, the NSP1 proteins inhibited IFN-λ1 expression to a greater extent than they inhibited IFN-β expression (Figure 1A). In equivalent experiments with the IFN-λ3 reporter, the RG177, BID2QJ and Wa NSP1 proteins reduced activation by 20–40% in two of the three experimental repeats; however, the lack of inhibition on the third occasion meant the result was not significant (Figure 4B). In contrast, P343, UKtc and particularly the RRV NSP1s were extremely effective inhibitors of the IFN-λ3 reporter, blocking activation by more than 80%. It has been reported that regulation of the IFN-λ3 promoter resembles induction of the IFN-α genes, in that it shows a greater dependence on IRF-7 than on IRF-3 and NF-κB [44]. Since RRV NSP1 had no effect on the ability of IRF-7 to stimulate the IFN-α4 promoter (Figure 2), it was therefore surprising to observe that it completely blocked activation of the IFN-λ3 promoter (Figure 4B). We considered the possibility that the effect of NSP1 on IRF-7 might be dependent on the promoter context, so we next tested whether NSP1 could block activation of the IFN-λ3 promoter by IRF-7 overexpression (Figure 4C). However, the results were very similar to those obtained using IRF-7 and the IFN-α4 promoter, with very little inhibition including by RRV NSP1. It is worth noting that P343, UKtc, and RRV NSP1s were stronger inhibitors of IRF-3 than the OSU-like NSP1s and the only ones to block the IFN-λ3 promoter efficiently, therefore the effects of NSP1 proteins on activation of IFN-λ3 may be primarily mediated through inhibition of IRF-3 rather than IRF-7.

### 3.6. RVA NSP1 Inhibits IRF-1

The IFN-β, IFN-λ1 and IFN-λ3 promoters have been shown to bind IRF-1 in addition to IRF-3 and IRF-7 [44,46]. IRF-1 is only expressed at very low levels in most cell types, however higher expression levels are observed in immune cells, and it is strongly induced by virus infection, IFN-α, and IFN-γ in other cells. Although IRF-1 is not required for primary IFN-β induction [47], it has been proposed that it plays an important role in induction of IFN-λ1 downstream of peroxisomal MAVS complexes [13]. Therefore, we decided to investigate if RVA NSP1 could inhibit activation of a reporter gene by transfected IRF-1. For this experiment we used the PRDIII-I element of the IFN-β promoter which is strongly activated by IRF-1 overexpression (Figure 5, black bars). Strikingly, we observed that all NSP1 proteins were very effective inhibitors of IRF-1 (Figure 5). Notably, the NSP1 proteins from the human RVA strains were markedly better inhibitors of IRF-1 than they were of IRF-3 or IRF-7. In contrast, the CSFV N^pro^ protein which served as a positive control for inhibition of IRF-3 and IRF-7, had no effect on IRF-1. Arnold et al. have previously reported that SA11-4F and WI61 NSP1 proteins promote degradation of IRF-3, IRF-7 and IRF-9, but have no effect on IRF-1 levels [20]. Consistent with this study, we also found no evidence that IRF-1 is degraded in cells expressing NSP1, suggesting that the mechanism of inhibition of IRF-1 is distinct from the mechanism of inhibition of IRF-3, IRF-7 and IRF-9.

### 3.7. Inhibition of IFN Signaling by RVA NSP1

Like many viruses, RVs interfere with multiple aspects of the innate immune response, and in addition to disrupting the production of IFNs by infected cells, they also block Jak/STAT signaling downstream of the type I IFN receptor. This is due to the ability of NSP1 to block both phosphorylation and nuclear import of activated STAT1/2 dimers, and to degrade IRF-9 [20,34,35,48]. Available data suggests that this property is conserved between all RVA strains. To examine inhibition of Jak/STAT signaling by the RVA NSP1 proteins studied here, we used a luciferase reporter gene under the control of the murine Mx1 promoter which was stimulated by treatment with IFN-α. The V protein from rinderpest virus (RPV-V) is a well characterized inhibitor of Jak/STAT signaling and served as a positive control (Figure 6, dark green bars). All RVA NSP1 proteins inhibited activation of the Mx1 promoter by IFN-α by at least 50% (Figure 6). Since type III IFN signaling also depends on the Jak/STAT pathway, it is likely that RV NSP1 similarly inhibits ISG expression downstream of the type III IFN receptor.

## 4. Discussion

The host IFN response is a powerful anti-viral mechanism that requires effective countermeasures by viruses to facilitate virus replication. A common, if not universal, strategy is the production of viral proteins that interact with components of the IFN induction and/or IFN signaling pathways to disrupt their function. Due to their limited coding capacity, viruses with RNA genomes often encode pleiotropic IFN antagonists with the ability to interact with several host proteins involved in both pathways. The genetic diversity exhibited by RVA NSP1 and its importance in host range restriction makes it of interest to investigate the interactions between NSP1 proteins and their cellular targets, and to explore how this might alter the early host response to infection. Here we studied inhibition of the human type I and type III IFN response by NSP1 proteins from a variety of human and animal RVA strains.

All NSP1 proteins examined inhibited induction of IFN-β in response to SeV, with the exception of ROTA04 NSP1 (Figure 1A). This group of NSP1s contained examples of proteins with either pLxIS or DSGIS motifs at their C-termini, but surprisingly all had the ability to inhibit both IRF-3 and NF-κB dependent transcription (Figure 1D and Figure 3A). P343, UKtc and RRV NSP1s were more effective inhibitors of IRF-3-5D than the NSP1s from the three human RVA strains and ROTA04 (Figure 1D), and in the yeast two-hybrid assay P343 and UKtc NSP1s interacted more strongly with IRF-3 than the other NSP1s (Figure 1E). These data show that although the presence of the pLxIS motif in NSP1 is associated with increased binding and inhibition of IRF-3, those NSP1 proteins without pLxIS may still bind IRF-3 and cause some reduction in activity. It is worth noting that truncations or point mutations in the N-terminal RING domain of NSP1 disrupt IRF-3 binding, indicating that the interaction is not solely dependent on the C-terminus [17,21], and OSU NSP1 bound weakly to IRF-3 in a GST-pulldown assay [21]. Many studies of NSP1 function have focused on the ability of NSP1 proteins from different strains to cause degradation of host factors rather than performing assays that measure the effect of NSP1 on transcriptional activity. Consequently, the possibility that some NSP1 proteins may interact with IRFs and inhibit their activity by mechanisms other than protein degradation may have been underappreciated.

Surprisingly, RRV NSP1 did not interact with IRF-3 in the yeast two-hybrid assay, and this was not due to species differences between human and simian IRF-3, as it also failed to interact with IRF-3 from the homologous host (Figure 1E). However, RRV NSP1 was as effective as P343 and UKtc NSP1s in the reporter assays, demonstrating that despite this apparent lack of binding it is fully functional as an IRF-3 antagonist. Since previous studies have shown that RRV NSP1 can degrade hIRF-3 [22], it is possible that there are technical reasons for failing to detect an interaction between RRV NSP1 and IRF-3 in yeast, such as incorrect folding or post-translational modifications that do not occur correctly in this system.

In general, all NSP1 proteins were less efficient inhibitors of IRF-7 than IRF-3, especially at high IRF-7 expression levels (Figure 2A). Whereas only UKtc NSP1 was able to reduce IRF-7-dependent transcription at high IRF-7 concentrations, RG177, Wa and P343 also showed some ability to inhibit IRF-7 in experiments using less IRF-7 plasmid (Figure 2B). Under the same conditions, CSFV N^pro^ which was used as a positive control, reduced reporter gene activity by 70% in the experiment with high levels of IRF-7, and this increased to >95% in the experiment with lower IRF-7 levels, demonstrating that a protein with a known ability to bind and inhibit IRF-7 [42] can block IRF-7-dependent transcription in this assay.

Phylogenetic analysis of more than 500 NSP1 sequences from a diverse range of hosts, showed that NSP1s from mammalian RVAs cluster into three groups, the OSU-like NSP1s which predominantly include the human and porcine RVA proteins, the UK-like NSP1s (including UKtc and P343), and the SA11-4F-like NSP1s (which includes RRV) [27]. The C-termini of the vast majority of OSU-like NSP1s contain the DSGIS motif characteristic of this group, however ROTA04 NSP1 is an exception to this rule. Despite lacking this sequence, ROTA04 showed no defect in inhibition of NF-κB activation by TNFα. In fact, all NSP1 proteins tested here showed a similar level of inhibition of NF-κB in response to β-TrCP overexpression regardless of the type of C-terminal motif present (Figure 3C). We attempted to establish whether NSP1 proteins can bind β-TrCP using the yeast two-hybrid assay; however, no interactions were observed. Since NSP1 needs to be phosphorylated in order to interact with β-TrCP, it may be impossible to detect the interaction if this modification does not occur in yeast. Interestingly OSU, UKtc, and RRV NSP1s have a modest effect on overexpressed p65, which suggests that they might have an additional mechanism to block NF-κB that is independent of β-TrCP, although this requires further investigation. It is worth noting that in experiments in which the C-terminal 12 amino acids of SA11-4F NSP1, which does not inhibit NF-κB activation, were replaced with the equivalent region of OSU NSP1, the chimera was able to efficiently degrade β-TrCP but was impaired in its ability to inhibit NF-κB with respect to intact OSU NSP1 [27]. This suggests that other parts of NSP1 may prevent activation of NF-κB by an alternative mechanism, which is consistent with our observations on p65 and inhibition of NF-κB by NSP1 proteins lacking DSGIS.

Type III IFNs play an important role in protecting against viral infections at mucosal surfaces such as the intestinal epithelium and are critical to controlling RV infection in vivo [8]. RV specifically replicates in the mature enterocytes at the tips of the villi lining the small intestine, and it has been shown that these cells switch from a type I to a type III IFN response as they mature and differentiate, associated with an increased content of peroxisomes [13]. Thus it was of interest to investigate whether RVs have the ability to counteract the production of type III IFNs. Similar to the effects observed on the IFN-β promoter, we found that all NSP1 proteins except ROTA04 inhibited IFN-λ1 induction (Figure 4A), and in general the inhibition was slightly more significant in the case of IFN-λ1. In contrast, we observed selective inhibition of IFN-λ3 expression by P343, UKtc and RRV NSP1s (Figure 4B), and although previous reports suggested that activation of the IFN-λ3 promoter relies on IRF-7 rather than IRF-3 [44], the effects of NSP1 proteins on IFN-λ3 did not correlate with their ability to inhibit IRF-7, especially in the case of RRV. Although the IFN-β, IFN-λ1 and IFN-λ3 promoters all bind IRF-1, IRF-3, IRF-7, and NF-κB p50/p65, the structures of the promoter regions are quite different [44,45]. In the IFN-β promoter, the IRF and NF-κB binding sites are immediately adjacent to each other and are accompanied by a binding site for ATF2/c-jun. The binding of these factors to the IFN-β promoter occurs in a co-operative manner to form the well characterized enhanceosome complex, in which synergy between transcription factors drives a high level of expression [49]. The IFN-λ1 and IFN-λ3 promoters are less well studied, but the IFN-λ1 promoter shows some similarity to the IFN-β promoter in the kinetics of induction and the proximity of the IRF and NF-κB binding sites, however it lacks a site for ATF2/c-jun. The IFN-λ3 promoter is different, with the NF-κB sites located downstream of the transcription start site [44]. In this context, it is possible that the IRFs and NF-κB act separately rather than co-operatively to activate transcription, which may explain the relatively lower level of inducibility of this promoter. Saxena et al. reported that type III IFN mRNAs were highly upregulated in RV infected HIEs, but that only very low levels of protein were produced, indicating that RVs can limit the IFN response both pre- and post-transcriptionally [10]. Interestingly, they observed higher IFN-λ1 levels in HIEs infected with a human RV than with RRV, which is consistent with our data showing that RRV NSP1 is a better inhibitor of IFN-λ1 induction than the human RV NSP1s.

This study also reveals a previously undescribed function of RVA NSP1 as a powerful inhibitor of IRF-1 (Figure 5). It is notable that the NSP1 proteins from the three human RVA strains had relatively modest effects on IRF-3 and IRF-7, but almost completely blocked IRF-1 function. In addition, the porcine ROTA04 NSP1 had a marginal effect on IRF-3, no effect on IRF-7, but was a strong inhibitor of IRF-1. Interestingly, NSP1 inhibition of IRF-1 appears to differ from inhibition of IRF-3 in several ways. Firstly, the magnitude of inhibition was far greater in the case of IRF-1, secondly there was no difference in the ability of NSP1 proteins with or without the pLxIS motif to block IRF-1 as there was with IRF-3-5D, and thirdly there was no degradation of IRF-1. Arnold et al. had previously shown that NSP1 does not degrade IRF-1 [20], however their study did not investigate whether NSP1 affected IRF-1-dependent transcription. These data suggest that the mechanism of inhibition of IRF-1 involves blocking IRF-1 function without decreasing protein levels and is, therefore, distinct from the mechanism previously described for inhibition of the other IRFs. Unlike IRF-3/7, IRF-1 is constitutively active and does not require phosphorylation to enter the nucleus. In most cell types IRF-1 is expressed at a low basal level but is rapidly induced by various stimuli including virus infection, IFN-α/β, and IFN-γ. Studies of the transcriptomes of IRF-1 knockout hepatocytes and respiratory epithelial cells demonstrated that IRF-1 is responsible for sustaining constitutive expression of a subset of ISGs that maintain a basal antiviral state in these tissues [50,51], and it is conceivable that IRF-1 has a similar role in the intestinal epithelium. This constitutive function, along with the critical role of IRF-1 in induction of type III IFNs, makes it advantageous for viruses that replicate at these sites to block IRF-1 activity. Indeed, porcine epidemic diarrhea virus (PEDV) which also replicates in IECs, has been shown to inhibit IRF-1-dependent type III IFN production [52]. Furthermore, a recent report demonstrated a role for IRF-1 in the upregulation of type III IFNs in human IECs in response to enterovirus infection, confirming the relevance of this transcription factor in the cells that support RV replication [53].

In summary, all NSP1s similarly inhibited NF-κB, IRF-1, and Jak/STAT signaling, but displayed differences in their relative abilities to antagonize IRF-3 and IRF-7. The OSU-like NSP1s were less effective inhibitors of IRF-3 when compared to the other NSP1s, and only some were weak inhibitors of IRF-7. The UK-like NSP1s were the most effective at inhibiting both IRF-3 and IRF-7, whilst the SA11-4F-like RRV NSP1 inhibited IRF-3 but not IRF-7. Although ROTA04 NSP1 had no defect in inhibition of NF-κB, it performed consistently less well than the human RV NSP1s in most other assays, particularly on the intact IFN-β and IFN-λ promoters. The combined effects of NSP1 on these transcription factors during infection will collectively influence the range of interferons and other cytokines present in the intestinal milieu, with a consequent impact on virus replication. Further studies into the specific interactions between NSP1 and host transcription factors will yield more insights into the complex interplay between virus and host.

## Figures and Tables

**Figure 1 viruses-13-00589-f001:**
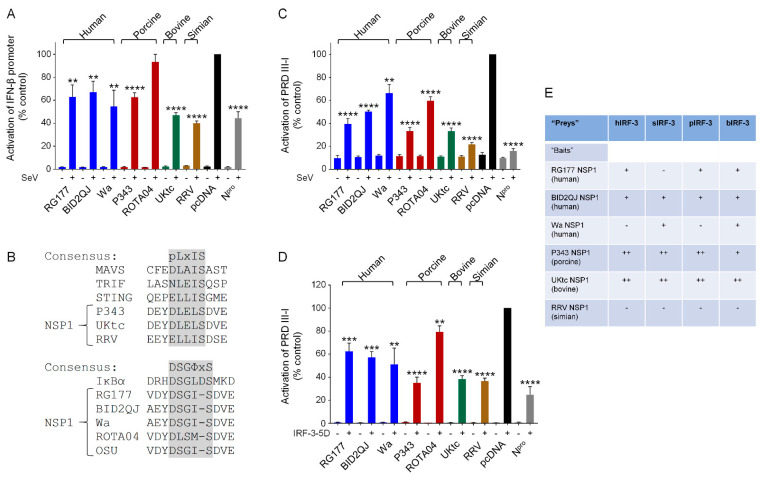
NSP1 (non-structural protein) from human and animal rotavirus strains inhibit interferon (IFN)-β induction by IFN regulatory factor (IRF)-3. (**A**,**C**,**D**) Human embryonic kidney (HEK)-293 cells were transfected with a plasmid containing a firefly luciferase gene under the control of the IFN-β promoter (**A**) or the PRDIII-I element from the IFN-β promoter (**C**,**D**), a plasmid directing constitutive expression of renilla luciferase as an internal control, and either the empty vector pcDNA3.1, or pcDNA3.1 encoding the NSP1 protein from human (RG177, BID2QJ, Wa), porcine (P343, ROTA04), bovine (UKtc) or simian (RRV) group A rotavirus (RVA) strains, or classical swine fever virus (CSFV) N^pro^ as a positive control. Reporter genes were activated by infection with Sendai virus (SeV) at 24 h post-transfection (**A**,**C**), or inclusion of a plasmid encoding constitutively active IRF-3 (IRF-3-5D) in the transfection (**D**). 48 h after transfection, cells were lysed and levels of firefly and renilla luciferase were measured. The normalized activity of the reporter gene in the samples transfected with the empty vector and induced by SeV (**A**,**C**) or IRF-3-5D (**D**) was set to 100%. Differences between induced samples containing NSP1 or N^pro^ and the induced control sample containing the empty vector were assessed using the Student’s *t*-test (** *p* < 0.01, *** *p* < 0.001, **** *p* < 0.0001). (**B**) Sequences of the C-terminal motifs present in different NSP1s. P343, UKtc, and RRV NSP1s contain a pLxIS motif (where p represents a hydrophilic residue and x is any amino acid) similar to the IRF-3 binding sites present in mitochondrial antiviral signaling protein (MAVS), TRIF, and STING. RG177, BID2QJ, Wa, and OSU NSP1s contain a DSGΦxS motif similar to that found in IκB (where Φ represents a hydrophobic residue and x is any or no amino acid). (**E**) Yeast transformed with plasmids encoding the NSP1 proteins from various RVA strains as “baits” and plasmids encoding human (h), simian (s), porcine (p), or bovine (b) IRF-3 as “preys” were streaked onto triple drop-out (TDO) and quadruple drop-out (QDO) media to assay for protein-protein interactions. ‘+’ indicates growth on TDO media but no growth on QDO media, ‘++’ indicates growth on QDO media, and ‘−‘ indicates no growth on either selection medium (i.e., no interaction).

**Figure 2 viruses-13-00589-f002:**
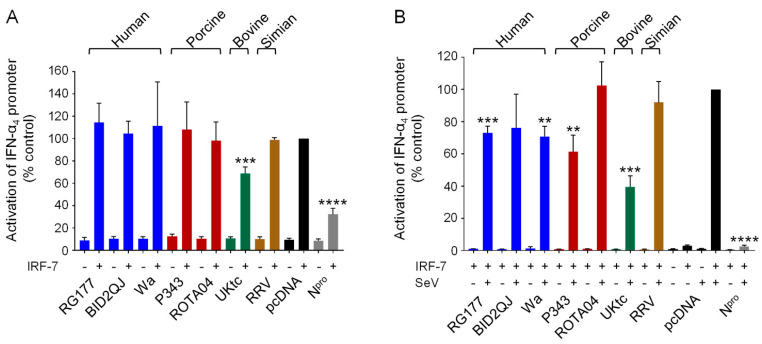
Inhibition of IFN-α4 induction by RVA NSP1s. (**A**,**B**) HEK-293 cells were transfected with a plasmid containing a firefly luciferase gene under the control of the IFN-α4 promoter, a plasmid directing constitutive expression of renilla luciferase as an internal control, a plasmid encoding IRF-7 where indicated, and either the empty vector pcDNA3.1, or pcDNA3.1 encoding the NSP1 protein from the indicated RVA strains, or CSFV N^pro^ as a positive control. Where indicated, cells were infected with SeV at 24 h post-transfection (**B**). After a further 16h, cells were lysed and levels of firefly and renilla luciferase were measured. The normalized activity of the reporter gene in the samples transfected with the empty vector and induced by IRF-7 (**A**) or IRF-7 + SeV (**B**) was set to 100%. Differences between induced samples containing NSP1 or N^pro^ and the induced control sample containing the empty vector were assessed using the Student’s *t*-test (** *p* < 0.01, *** *p* < 0.001, **** *p* < 0.0001).

**Figure 3 viruses-13-00589-f003:**
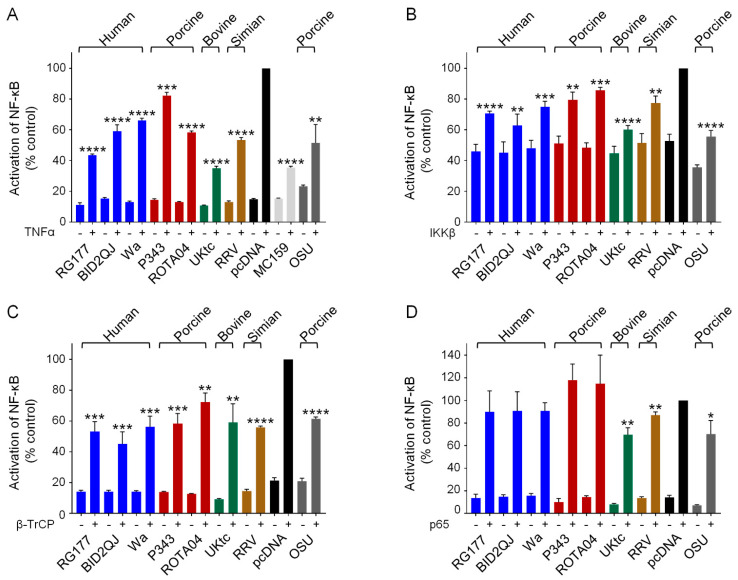
Inhibition of NF-κB activation by RVA NSP1s. (**A**–**D**) HEK-293 cells were transfected with a plasmid containing a firefly luciferase gene under the control of an NF-κB-responsive promoter, a plasmid directing constitutive expression of renilla luciferase as an internal control, either the empty vector pcDNA3.1, or pcDNA3.1 encoding the NSP1 protein from the indicated RVA strains, or MCV MC159 as a positive control. Reporter genes were activated by treatment with 5 ng/mL TNFα at 24 h post-transfection (**A**), or inclusion of a plasmid encoding IKKβ (**B**), β-TrCP (**C**) or p65 (**D**) in the transfection. 48 h after transfection, cells were lysed and levels of firefly and renilla luciferase were measured. The normalized activity of the reporter gene in the samples transfected with the empty vector and induced by TNFα (**A**), IKKβ (**B**), β-TrCP (**C**) or p65 (**D**) was set to 100%. Differences between induced samples containing NSP1 or MC159 and the induced control sample containing the empty vector were assessed using the Student’s *t*-test (* *p* < 0.05, ** *p* < 0.01, *** *p* < 0.001, **** *p* < 0.0001).

**Figure 4 viruses-13-00589-f004:**
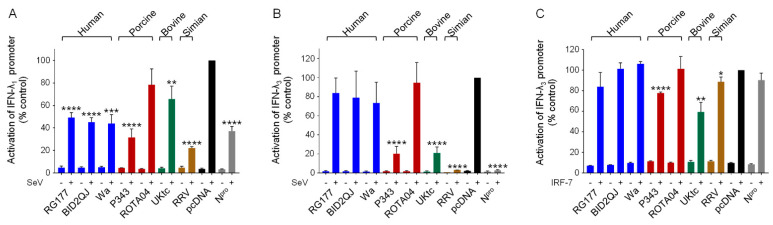
Inhibition of type III IFN induction by RVA NSP1s. (**A**–**C**) HEK-293 cells were transfected with a plasmid containing a firefly luciferase gene under the control of the IFN-λ1 promoter (**A**) or the IFN-λ3 promoter (**B**,**C**), a plasmid directing constitutive expression of renilla luciferase as an internal control, a plasmid encoding IRF-7 where indicated, and either the empty vector pcDNA3.1, or pcDNA3.1 encoding the NSP1 protein from the indicated RVA strains, or CSFV N^pro^ as a positive control. Where indicated, cells were infected with SeV at 24 h post-transfection. After a further 16 h, cells were lysed and levels of firefly and renilla luciferase were measured. The normalized activity of the reporter gene in the samples transfected with the empty vector and induced by SeV was set to 100%. Differences between induced samples containing NSP1 or N^pro^ and the induced control sample containing the empty vector were assessed using the Student’s *t*-test (* *p* < 0.05, ** *p* < 0.01, *** *p* < 0.001, **** *p* < 0.0001).

**Figure 5 viruses-13-00589-f005:**
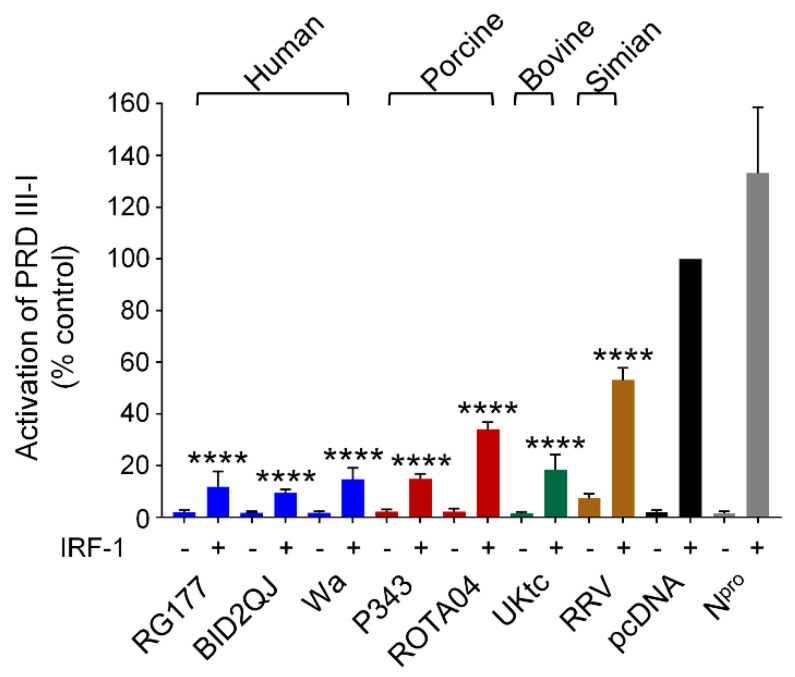
RVA NSP1 inhibits IRF-1. HEK-293 cells were transfected with a plasmid containing a firefly luciferase gene under the control of the PRDIII-I element from the IFN-β promoter, a plasmid directing constitutive expression of renilla luciferase as an internal control, a plasmid encoding IRF-1 where indicated, and either the empty vector pcDNA3.1, or pcDNA3.1 encoding the NSP1 protein from the indicated RVA strains, or CSFV N^pro^; 48 h after transfection, cells were lysed and levels of firefly and renilla luciferase were measured. The normalised activity of the reporter gene in the sample transfected with the empty vector and IRF-1 was set to 100%. Differences between induced samples containing NSP1 and the induced control sample containing the empty vector were assessed using the Student’s *t*-test (**** *p* < 0.0001).

**Figure 6 viruses-13-00589-f006:**
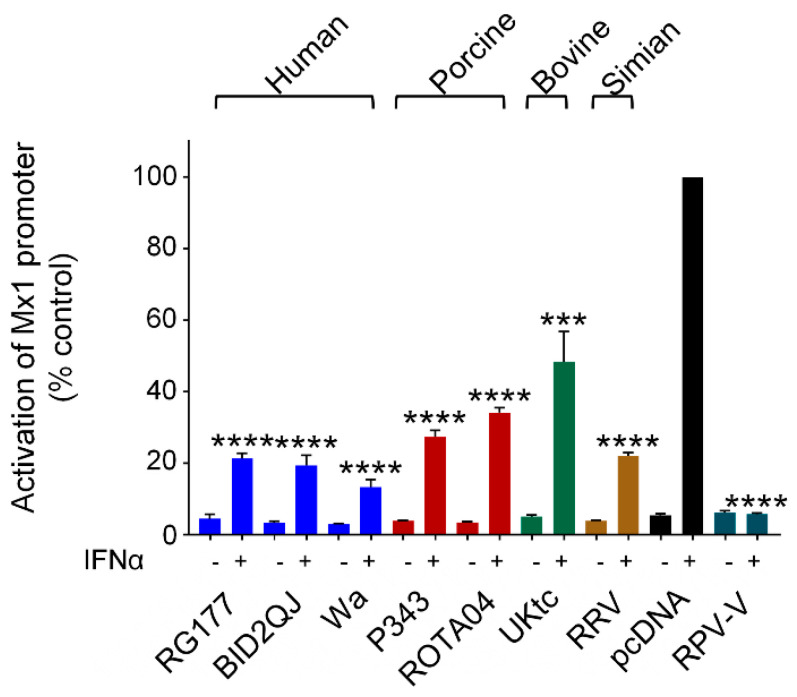
Inhibition of IFN signaling by RVA NSP1. HEK-293 cells were transfected with a plasmid containing a firefly luciferase gene under the control of the Mx1 promoter, a plasmid directing constitutive expression of renilla luciferase as an internal control, and either the empty vector pcDNA3.1, or pcDNA3.1 encoding the NSP1 protein from the indicated RVA strains, or the V protein from RPV as a positive control. Where indicated, cells were treated with 500 IU/mL IFN-α 24 h after transfection. After a further 16h, cells were lysed and levels of firefly and renilla luciferase were measured. The normalised activity of the reporter gene in the sample transfected with the empty vector and treated with IFN-α was set to 100%. Differences between induced samples containing NSP1 and the induced control sample containing the empty vector were assessed using the Student’s *t*-test (*** *p* < 0.001, **** *p* < 0.0001).

## Data Availability

The authors confirm the availability of the data supporting this study.

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
