# Peer review of "Rotavirus NSP1 Inhibits Type I and Type III Interferon Induction"

_viruses, 2021, doi:10.3390/v13040589_

Round 1

Reviewer 1 Report

This article by Iaconis et al using in vitro cell culture models demonstrates that rotavirus NSP1 inhibits type I and type III interferon induction.

1) It is mentioned that the figures present data from 3 different experiments, but it is unclear if each experiment was done in duplicates or triplicates.
2) Using non transformed human intestinal enteroids, Saxena et al demonstrated that HRV replication was not restricted due to the endogenous response as replication-competent HRV antagonized the type III IFN response at pre- and posttranscriptional levels. On the contrary, it was the exogenous IFN treatment that restricted HRV replication and type I IFN was more potent than type III IFN.  The authors should discuss their study results in light of this article as well (Saxena et al. PNAS 2017, A paradox of transcriptional and functional innate interferon responses of human intestinal enteroids to enteric virus infection).

Author Response

Dear Reviewers

Thank you for taking the time to review and for your comments and feedback, which we feel have further improved the manuscript.

Kind regards

Dr Julian Seago

Responses to reviewer’s comments:

Reviewer 1:

1) It is mentioned that the figures present data from 3 different experiments, but it is unclear if each experiment was done in duplicates or triplicates.

Each experiment was done once on 3 separate occasions. The following changes were made to the text to clarify this and reduce repetition:

The phrase “Experiments were repeated 3 times and the data presented are the mean with the error bars representing the standard deviation.” has been removed from the legends to figures 1-6 and this has been replaced with “For all experiments, the data presented are the means of 3 readings obtained from 3 independent experiments conducted on separate occasions, with the error bars representing the standard deviation.” in the materials and methods section 2.4.

2) Using non transformed human intestinal enteroids, Saxena et al demonstrated that HRV replication was not restricted due to the endogenous response as replication-competent HRV antagonized the type III IFN response at pre- and posttranscriptional levels. On the contrary, it was the exogenous IFN treatment that restricted HRV replication and type I IFN was more potent than type III IFN.  The authors should discuss their study results in light of this article as well (Saxena et al. PNAS 2017, A paradox of transcriptional and functional innate interferon responses of human intestinal enteroids to enteric virus infection).

These data have been briefly mentioned in the introduction and related to our findings in the discussion. This reference has been inserted as reference number 10, and the numbering of existing references has been adjusted accordingly.

Reviewer 2:

It slightly dampens the enthusiasm regarding this manuscript that the novelty of these findings is somewhat limited (or not emphasized well). The observations that RVs antagonize production of type I and III IFNs via degradation of several IRFs by NSP1 were previously published and broadly acknowledged. The authors should do a better job emphasizing their research rationale and what is novel about their findings instead of currently used vague statements that this information is of importance.

Changes have been made to the abstract to emphasize the novel aspects of this work.

The observation that NSP1 degrades IRF1 is novel, but it contradicts the previous observations that NSP1 does not degrade IRF1 because it lacks IAD (Arnold et al., 2013). This needs to be acknowledged and this discrepancy should be discussed.

We do not show that NSP1 degrades IRF-1, in fact we state (L510 in track-changed document) that we saw no evidence of IRF-1 degradation in our experiments which is consistent with the results of Arnold et al. Our data show that NSP1 blocks transcriptional activation by IRF-1 through a distinct mechanism that does not involve protein degradation. Changes have been made to the text to emphasize the differences in our two studies (L683 in track-changed document).

Other comments:

P1, L35-36: “RVs can be further differentiated into 35 at least 8 distinct species, named RVA-RVH,” RVs are currently classified into 10 species RVA-RVJ, or 9 if omit RVEs. Not clear why it is stated ‘at least 8’ and ‘RVA-RVH’, and not ‘at least 9’ and ‘RVA-RVJ’. Please revise.

The text (now L38 in track-changed document) has been revised.

The manuscript is unnecessarily lengthy and should be significantly condensed by: a) shortening Introduction and Discussion and b) using more concise language and avoiding being redundant in Results. For example, P6, L248-257 should be omitted – these are not results, these are methods already mentioned in M&Ms and detailed in each respective Figure legends. Similar modifications need to be done throughout the Results section.

The text has been modified throughout to be more concise and avoid repetition.

List of changes made (Line numbers refer to track changed document)

L10        deleted “SW19” and inserted “SW17” to correct a typographical error in the postcode for affiliation 3.

Abstract

L24        inserted “provide the first demonstration” and deleted “demonstrate”.

L26        inserted “We also show” and deleted “Furthermore, we found”

L27        inserted “ a previously undescribed property of NSP1 which is conserved among human and animal RVs.”

L27-8     deleted “and it was notable that “ and inserted “Interestingly, “.

L29        deleted “. This is likely important” and inserted “which has significance”.

Introduction

L35        inserted “ that are grouped”

L35-8     deleted “(reviewed in [1]). The RV genome consists of 11 double-stranded RNA (dsRNA) segments encoding 12 proteins, and is encapsidated in a triple-layered, non-enveloped virion with icosahedral symmetry. RVs can be further differentiated “

L38        deleted “8” and inserted “9”

L38        deleted “RVH” and inserted “RVJ (reviewed in [1]).”

L38-9     deleted “,with the group “ and inserted “Group”             

L39-40 deleted “constituting important medical and veterinary pathogens. RVs are transmitted via the fecal-oral route and “

L41-3     inserted “ and are “ and deleted “, causing severe dehydrating diarrhea with significant associated morbidity and mortality worldwide. The recent introduction of live-attenuated vaccines has reduced the number of fatalities considerably, however RVAs remain “

L47        deleted “have shown ”

L47        inserted “showed ”

L59        deleted “restricting “

L60        deleted “binding to “

L66        deleted “it has been shown that “

L70-1     deleted “.” and inserted “, a feature that was also observed in RV-infected human intestinal enteroids (HIEs) [10]. Furthermore, s” and deleted “S”.

L71        deleted “have shown “

L72        inserted “showed ”

L76        deleted “.” and inserted “, and interestingly, exogenous type I IFN was more effective than type III in controlling RV infection of HIEs [10].”

L78        deleted “type ”

L80        added “.” And deleted “such as the toll-like receptors (TLRs) or RIG-I-like receptors (RLRs). “

L83        deleted “10” and “11” and inserted “11” and “12”

L84        deleted” located on the outer surface of mitochondria and peroxisomes, ”

L88        deleted “which enables “ and inserted “enabling ”

L88        deleted “IRF-3 “

L88        inserted “nuclear “

L89        deleted “ into the nucleus”

L92-3     deleted “. Loss of the inhibitor protein releases” and inserted “releasing “

L93        inserted “ to” and deleted “, which can then”

L94        deleted “promoters of the “

L95        inserted “promoters” and deleted “genes”

L98        deleted “One intriguing report suggested that whilst m”, added “M”

L100      changed “12” to “13”

L101-2  changed “13” to “14”, deleted “with evidence that IFN-β can in fact be induced via peroxisomal MAVS ”

L106      changed “14” to “15”

L106-7  inserted “bind “ and deleted “form a trimolecular complex (ISGF3) with the DNA binding protein “

L107      inserted “ to form ISGF3”

L108-9  deleted “The products of these ISGs have a wide range of anti-viral activities which limit the infection. ”

L110      deleted “ in the presence of a functional IFN system”

L113      deleted “has been shown to “ and inserted “s”

L115      changed “15” to “16”

L123      changed “16” to “17” and “19” to “20”

L125      deleted “IRF-3-dependent “ and changed “16” to “17”, “17” to “18” and “20” to “21”.

L128      changed “21” to “22”

L130      changed “22” to “23”

L132      changed “23” to “24”, inserted “,” and deleted “. β-TrCP is “

L134      deleted “ubiquitination and “ and changed “24” to “25”.

L136      deleted “is a property that “

L137      changed “25” to “26”

L138-9  deleted “, and sequence differences in this region determine target selection”

L139      changed “16” to “17”, “25” to “26” and “26” to “27”.

L141      deleted “which is “

L143      deleted “for subsequent activation “

L143      changed “27” to “28”

L145      changed “28” to “29”

L146      changed “27” to “28”

L148      changed “25” to “26” and “26” to “27”

L148-9  inserted “, where Φ represents a hydrophobic residue”

L153      changed “29” to “30”

L155      changed “30” to “31”, “31” to “32” and “32” to “33”

L157      changed “33” to “34” and “34” to “35”

L159-60                             inserted “investigated whether RV NSP1 proteins inhibit the production of type III in addition to type I IFNs using”

L160      deleted “compare the ability of NSP1 proteins from “

L161      inserted “well characterized and previously untested NSP1 proteins from “

L162      deleted “ for their ability to antagonize the IFN response in human cells”

L163      inserted “activation of “ and deleted “RV NSP1 can inhibit “

L163-4  inserted “by virus infection is inhibited by human and animal RV NSP1s” and deleted “as well as type I”

Materials and methods

L177      inserted “accession number (AN)” and deleted “AN”

L181-2  deleted “accession number (AN)” and inserted “AN: ”

L201      changed “35” to “36” and “36” to “37”

L202      changed “37” to “38”

L215      changed “38” to “39”

L216      changed “39” to “40”

L250-3  inserted “For all experiments, the data presented are the means of 3 readings obtained from 3 independent experiments conducted on separate occasions, with the error bars representing the standard deviation.”

Results

L262      inserted “First we”, deleted “We set out to”and inserted “d”

L263-4  inserted “SeV “ and “a luciferase reporter gene under the control of “ and deleted “human “

L264-71               deleted “cDNAs encoding the NSP1 proteins from three human RVA strains (RG177, BID2QJ and Wa), two porcine RVA strains (P343 and ROTA04), one bovine RVA strain (UKtc), and one simian RVA strain (RRV) were cloned into the expression vector pcDNA3.1, and transfected into HEK-293 cells along with a plasmid containing a firefly luciferase gene under the control of the human IFN-β promoter and a renilla luciferase expression plasmid as an internal transfection control. To stimulate the IFN-β reporter, cells were infected with Sendai virus (SeV) which”

L271      inserted “SeV infection “

L278      changed “35” to “36” and “40” to “41”

L288-9  deleted “Experiments were repeated 3 times and the data presented are the mean with the error bars representing the standard deviation.”

L301      inserted “on ” and deleted “upon activation of “

L308      changed “26” to “27”

L323-4  inserted “so in “ and deleted “but IRF-3 is primarily responsible for activation by SeV in HEK-293 cells. In ”

L326      inserted “inducibly phosphorylated “

L327      deleted “that are inducibly phosphorylated in response to virus infection, “

L328      deleted “the “

L336-7  deleted “ even though they lack the previously characterized IRF-3 binding site”

L338      changed “21” to “22”

L339      deleted “the “

L340-1  deleted “different “

L348      deleted “at the C-terminus “

L350      inserted “luciferase “ and deleted “reporter gene “

L354      deleted “which suggests “ and inserted “suggesting “

L355      deleted “perhaps “

L357      inserted “luciferase “ and deleted “reporter gene “

L359      deleted “reporter “ and inserted “luciferase “

L361      deleted “also “ and “to “ and inserted “ either”

L362      inserted “it “ and deleted “this NSP1 “ and inserted “perhaps “

L363      deleted “it is possible that it “

L364      inserted “is no “

L364-5  deleted “does not appear to be any “ and inserted “s”

L369      deleted “for degradation “

L371      changed “18” to “19”, “19” to “20” and “22” to “23”

L377      changed “41” to “42”

L380      deleted “much “

L393      deleted “Experiments were repeated 3 times and the data presented are the mean with the error bars representing the standard deviation.”

L398-9  deleted “The data presented in section 3.2 show that NSP1 proteins with both types of C-terminal motif can inhibit IRF-3-dependent transcription in a reporter gene assay. ”

L403      changed “42” to “43”

L404-5  deleted “which contains a DSGIS motif and degrades β-TrCP “ and changed “23” to “24”

L408-9  deleted “the “ and “motif “

L418      deleted “Experiments were repeated 3 times and the data presented are the mean with the error bars representing the standard deviation.”

L422-4  deleted “To test whether the NSP1 proteins interact with β-TrCP, we performed yeast two-hybrid assays with the various NSP1s as baits and β-TrCP as the prey. However, no interactions between β-TrCP and any NSP1 protein were observed (data not shown). “

L425      deleted “the “

L431      deleted “ containing IKKα, IKKβ and IKKγ/NEMO”

L432      deleted “the “ and “complex “

L445      deleted “ here”

L451-3  deleted “Type III IFNs have a specific role in antiviral defense at mucosal surfaces such as the intestinal epithelium, and studies in receptor knockout mice indicate that type III rather than type I IFNs are critical for the control of RV infections in vivo [8]. ”

L455      changed “43” to “44” and “44” to “45”

L456-7  inserted “, which may be critical to countering the innate immune response in the intestinal epithelium”

L461-2  added “Except for UKtc NSP1”, deleted “In general, the NSP1 proteins, with the exception of UKtc NSP1,”

L464      deleted “from the human RVA strains “

L470      changed “43” to “44”

L490-1  deleted “Experiments were repeated 3 times and the data presented are the mean with the error bars representing the standard deviation.”

L496      changed “43” to “44” and “45” to “46”

L499      changed “46” to “47”

L501      changed “12” to “13”

L510      changed “19” to “20”

L521-2  deleted “Experiments were repeated 3 times and the data presented are the mean with the error bars representing the standard deviation.”

L530      changed “19” to “20”, “33” to “34”, “34” to “35” and “47” to “48”

L547-8  deleted “Experiments were repeated 3 times and the data presented are the mean with the error bars representing the standard deviation.”

Discussion

L553      inserted “facilitate” and deleted “create an environment that is favourable for”

L553      deleted “in order”

L557      deleted “such as the RVA NSP1 protein, “

L558      deleted “and inhibit “ and “independent “

L559      inserted “The genetic diversity exhibited by RVA “

L559-60 deleted “shows notable genetic diversity and has been implicated”

L560      inserted “and it’s importance” and “ makes it of “

L561      deleted “. It is therefore of considerable “

L565      deleted “the “ and “here “

L569      deleted “noticeably “

L573      inserted “and inhibition of “ and deleted “to “

L573-4  deleted “ and increased inhibition of IRF-3”

L575      deleted “IRF-3 “ and “In this regard it is also “ and inserted “It is “

L576-8  inserted “ truncations or point mutations in the N-terminal RING domain of NSP1 disrupt IRF-3 binding, indicating that the interaction is not solely dependent on the C-terminus [17,21], and”

L578      deleted “, which is the prototypical β-TrCP-targeting NSP1 protein,”

L579      changed “20” to “21” and inserted “.”

L579-83               deleted “, although we did not observe any inhibition of IRF-3-5D activity in the presence of OSU NSP1 (data not shown). Further evidence that other parts of the protein outside of the C-terminal motif may be involved in IRF-3 binding, comes from experiments with N-terminal truncations of NSP1 and NSP1 containing point mutations in the RING domain, which fail to bind IRF-3 [16,20].”

L594      changed “21” to “22”

L605      changed “41” to “42”

L605      deleted “is able to”, added “can”

L611      changed “26” to “27”

L616      deleted “both types of “ and inserted “s”

L617-8  inserted “no interactions were observed.”

L618      deleted “none of the NSP1 proteins interacted with β-TrCP in this system.”

L619-20 deleted “if this modification does not occur in yeast “

L620-1  inserted “if this modification does not occur in yeast”

L621      deleted “using this technique”

L627      changed “26” to “27”

L631      inserted “play” and deleted “have”

L631      deleted “providing protection “ and inserted “protecting “

L632      deleted “respiratory and “

L632-8  inserted “ epithelium, and are critical to controlling RV infection in vivo [8]. RV specifically replicates in the mature enterocytes at the tips of the villi lining the small intestine, and it has been shown that these cells switch from a type I to a type III IFN response as they mature and differentiate, associated with an increased content of peroxisomes [13]. Thus it was of in-terest to investigate whether RVs have the ability to counteract the production of type III IFNs.”

L638-47 deleted “epithelia. This specificity is driven by the restricted expression of the type III IFN receptor subunit IFNLR1, which is found exclusively on epithelial cells and some immune cells. RV replicates in the mature enterocytes at the tips of the villi lining the small intestine, and it has been shown that these cells produce and respond to type III IFNs [8]. Interestingly, it also appears that IECs switch from a type I to a type III IFN response as they mature and differentiate, and this may be associated with an increased content of peroxisomes [12]. Experiments in mice lacking functional receptors for either type I or type III IFNs have revealed that type III IFNs have a critical, non-redundant role in controlling RV infection in vivo [8], thus it was of interest to investigate whether RVs have the ability to counteract the production of type III IFNs. “

L651      deleted “have “

L652      changed “43” to “44”

L655      deleted “contain binding sites for IRFs and NF-κB, and have been shown to “

L657      changed “43” to “44” and “44” to “45”

L661      changed “48” to “49”

L665      changed “43” to “44”

L667-73               inserted “ Saxena et al reported that type III IFN mRNAs were highly upregulated in RV infected HIEs, but that only very low levels of protein were produced, indicating that RVs can limit the IFN response both pre- and post-transcriptionally [10]. Interestingly, they observed higher IFN-λ1 levels in HIEs infected with a human RV than with RRV, which is consistent with our data showing that RRV NSP1 is a better inhibitor of IFN-λ1 induction than the human RV NSP1s.”

L671      deleted “most “

L683-85 inserted “Arnold et al had previously shown that NSP1 does not degrade IRF-1 [20], however their study did not investigate whether NSP1 affected IRF-1-dependent transcription.”

L685      deleted “ This”

L685      inserted “These data “ and “s”

L685-6  inserted “involves blocking IRF-1 function without decreasing protein levels, and “

L686      inserted “therefore “

L687      inserted “the mechanism”

L687      deleted “that”

L691      deleted “have “

L693      changed “49” to “50” and “50” to “51”

L698      changed “51” to “52”

L701      changed “52” to “53”

L702      deleted “had a conserved ability to “ and inserted “similarly “ and “ed”

References

L749-51 inserted reference 10 “Saxena, K.; Simon, L.M.; Zeng, X-L.; Blutt, S.E.; Crawford, S.E.; Sastri, N.P.; Karandikar, U.C.; Ajami, N.J.; Zachos, N.C.; Kovbasnjuk, O.; et al. A paradox of transcriptional and functional innate interferon responses of human intestinal enteroids to enteric virus infection. Proc. Natl. Acad. Sci. U S A. 2017, 114, E570-E579, doi:10.1073/pnas.1615422114.”

L752-852            changed reference numbers from 11 upwards.

Reviewer 2 Report

In this manuscript, the authors demonstrate that NSP1 proteins from several human and animal RV strains antagonize type I and III IFN induction and inhibits IRF-1 more efficiently than either IRF-3 or IRF-7. These are important findings contributing to better understanding of immune evasive mechanisms utilized by RVs.

It slightly dampens the enthusiasm regarding this manuscript that the novelty of these findings is somewhat limited (or not emphasized well). The observations that RVs antagonize production of type I and III IFNs via degradation of several IRFs by NSP1 were previously published and broadly acknowledged. The authors should do a better job emphasizing their research rationale and what is novel about their findings instead of currently used vague statements that this information is of importance.

The observation that NSP1 degrades IRF1 is novel, but it contradicts the previous observations that NSP1 does not degrade IRF1 because it lacks IAD (Arnold et al., 2013). This needs to be acknowledged and this discrepancy should be discussed.

Other comments:

P1, L35-36: “RVs can be further differentiated into 35 at least 8 distinct species, named RVA-RVH,” RVs are currently classified into 10 species RVA-RVJ, or 9 if omit RVEs. Not clear why it is stated ‘at least 8’ and ‘RVA-RVH’, and not ‘at least 9’ and ‘RVA-RVJ’. Please revise.

The manuscript is unnecessarily lengthy and should be significantly condensed by: a) shortening Introduction and Discussion and b) using more concise language and avoiding being redundant in Results. For example, P6, L248-257 should be omitted – these are not results, these are methods already mentioned in M&Ms and detailed in each respective Figure legends. Similar modifications need to be done throughout the Results section.

Author Response

(The authors gave the same response as above.)
